# Feasibility of Community Pharmacist-Initiated and Point-of-Care *CYP2C19* Genotype-Guided De-Escalation of Oral P2Y12 Inhibitors

**DOI:** 10.3390/genes14030578

**Published:** 2023-02-25

**Authors:** Amar D. Levens, Melina C. den Haan, J. Wouter Jukema, Mette Heringa, Wilbert B. van den Hout, Dirk Jan A. R. Moes, Jesse J. Swen

**Affiliations:** 1Department of Clinical Pharmacy & Toxicology, Leiden University Medical Center, 2333 ZA Leiden, The Netherlands; 2Department of Cardiology, Leiden University Medical Center, 2333 ZA Leiden, The Netherlands; 3Netherlands Heart Institute, 3511 EP Utrecht, The Netherlands; 4SIR Institute for Pharmacy Practice and Policy, 2331 JE Leiden, The Netherlands; 5Department of Biomedical Data Sciences, Leiden University Medical Center, 2333 ZA Leiden, The Netherlands

**Keywords:** *CYP2C19*, pharmacogenetics, pharmacogenomics, P2Y_12_ inhibitor, clopidogrel, personalized medicine, point-of-care testing, dual antiplatelet therapy

## Abstract

Tailoring antiplatelet therapy based on *CYP2C19* pharmacogenetic (PGx) testing can improve cardiovascular outcomes and potentially reduce healthcare costs in patients on a P2Y_12_-inhibitor regime with prasugrel or ticagrelor. However, ubiquitous adoption—particularly in an outpatient setting—remains limited. We conducted a proof-of-concept study to evaluate the feasibility of *CYP2C19*-guided de-escalation of prasugrel/ticagrelor to clopidogrel through point-of-care (POC) PGx testing in the community pharmacy. Multiple feasibility outcomes were assessed. Overall, 144 patients underwent *CYP2C19* PGx testing in 27 community pharmacies. Successful test results were obtained in 142 patients (98.6%). De-escalation to clopidogrel occurred in 19 patients (20%) out of 95 (67%) eligible for therapy de-escalation, which was mainly due to PGx testing not being included in cardiology guidelines. Out of the 119 patients (84%) and 14 pharmacists (100%) surveyed, 109 patients (92%) found the community pharmacy a suitable location for PGx testing, and the majority of pharmacists (86%) thought it has added value. Net costs due to PGx testing were estimated at €43 per patient, which could be reduced by earlier testing and could turn into savings if de-escalation would double to 40%. Although the observed de-escalation rate was low, POC *CYP2C19*-guided de-escalation to clopidogrel appears feasible in a community pharmacy setting.

## 1. Introduction

Dual antiplatelet therapy (DAPT) with aspirin and a P2Y_12_ inhibitor is the standard of care among patients with an acute coronary syndrome (ACS) or stable coronary artery disease undergoing a percutaneous coronary intervention (PCI) [1]. Data suggest that ticagrelor and prasugrel are more potent antiplatelet drugs compared to clopidogrel [2]. However, the use of these potent P2Y_12_ inhibitors lowers the risk of ischemic events at the expense of an increased bleeding risk, lower compliance and higher drug expenses [3,4,5]. Clopidogrel remains the most commonly used and least expensive P2Y_12_ inhibitor [6,7,8]. It is a pro-drug that needs to be activated by the polymorphic cytochrome P450 *CYP2C19* enzyme. The best characterized loss-of-function (LOF) variant alleles are *CYP2C19**2 and *3. Both result in an impaired activity of *CYP2C19* enzymes and subsequent lower hepatic biotransformation of clopidogrel into its active metabolite. In 2009, the FDA published a boxed warning concerning the diminished antiplatelet effect in patients with a reduced *CYP2C19* function [9]. Clopidogrel treated-patients with one *CYP2C19* LOF allele have a higher risk of cardiovascular ischemic events in the post-myocardial infarction setting [10,11,12,13].

In the POPular Genetics trial, a *CYP2C19* genotype-guided P2Y_12_ inhibitor therapy in patients undergoing primary PCI was shown to be non-inferior compared to standard treatment with ticagrelor or prasugrel with respect to thrombotic events, and it resulted in a lower bleeding incidence [14]. Additionally, the TAILOR-PCI trial showed that a *CYP2C19* genotyped-guided treatment algorithm resulted in a similar efficacy compared to conventional treatment with ticagrelor in the POPular Genetics Trial [15]. Both the Clinical Pharmacogenomics Implementation Consortium (CPIC) and the Dutch Pharmacogenetics Working Group (DPWG) recommend in their guidelines to consider the use of alternative therapy for clopidogrel in patients carrying one or two *CYP2C19* LOF alleles [16,17]. The DPWG also considers *CYP2C19* genotyping prior to prescribing clopidogrel essential [18]. Furthermore, recent evidence shows that de-escalation of oral P2Y_12_ inhibitors therapy by switching from ticagrelor or prasugrel to clopidogrel is associated with a lower bleeding risk without an increased risk of ischemia [19,20,21]. However, despite the available evidence and guidelines, *CYP2C19* genotyping at the start of P2Y_12_ inhibitor treatment is not yet standard of care.

Community pharmacists, being among the most accessible healthcare professionals for patients, are considered ideal candidates to deliver pharmacogenetics to patient care [22,23,24,25]. This is further supported by the availability of point-of-care (POC) *CYP2C19* pharmacogenetic (PGx) tests and their current successful application for P2Y_12_ inhibitor de-escalation in the hospital setting [26]. Building on this success, we investigated whether *CYP2C19*-guided de-escalation from prasugrel or ticagrelor to clopidogrel in the outpatient setting by community pharmacists, using a POC *CYP2C19* device, would be feasible as well.

## 2. Materials and Methods

A multicenter proof-of-concept study was conducted to assess the feasibility of *CYP2C19*-guided de-escalation from prasugrel or ticagrelor to clopidogrel by community pharmacists using a POC *CYP2C19* device.

### 2.1. Study Setting

Pharmaceutical care in Dutch community pharmacy practice is held to high standards, as all pharmacies have electronic patient records, and medication surveillance and counseling are part of everyday routine practice. Up to 95% of the patients visit the same pharmacy and are argued to be well protected against many prescription drug-related problems [27].

Community pharmacies in the Netherlands were selected based on available patients on ticagrelor or prasugrel therapy at the time and willingness to participate. In total, 27 community pharmacies in three regions, comprising both independent and chain community pharmacies, participated in the study. Multiple community pharmacies had a cooperative relationship with each other. See Appendix A for full details.

### 2.2. Study Design

The primary aim of our study was to determine if it was feasible to implement POC *CYP2C19*-guided de-escalation from prasugrel or ticagrelor to clopidogrel by community pharmacists. Multiple feasibility outcomes were defined to assess the study aim.

The primary outcome was specified as the de-escalation rate of eligible study patients from ticagrelor or prasugrel to clopidogrel based on their *CYP2C19* genotype status. Eligible patients for de-escalation to clopidogrel were defined as patients without a *CYP2C19* LOF-allele (*CYP2C19**2/*CYP2C19**3). The second feasibility outcome was the success rate of the *CYP2C19* POC device, which was measured as the proportion of patients with a successful *CYP2C19* test result. The turnaround time of PGx testing was also defined as a feasibility outcome. Other outcomes included patient acceptance, comprehension and satisfaction, pharmacists’ perceived barriers, and facilitators and prescriber acceptance, and were they assessed by conducting patient-, pharmacist- and cardiologist-specific surveys. We also aimed to evaluate the potential cost savings of POC *CYP2C19*-guided DAPT de-escalation to clopidogrel in comparison to conventional DAPT treatment with ticagrelor/prasugrel.

### 2.3. Ethical Approval

All patients provided written informed consent prior to study enrollment. The study was approved by the Medical Ethical Committee Leiden-Den Haag-Delft (METC-LDD) (reference: N21.119) and conducted in accordance with the Declaration of Helsinki.

### 2.4. Study Population

Patients of the participating pharmacies were eligible to participate in this study. To be included in this study, adult patients (≥18 years) must currently be on a P2Y_12_ inhibitor-based antiplatelet therapy with either ticagrelor or prasugrel and have not been previously genotyped for *CYP2C19* variants. For the purposes of this feasibility study, patients had to have an indication for ticagrelor/prasugrel therapy for at least 3 more months so that a short remaining treatment could not affect the de-escalation rate. The exclusion criteria included patients with a documented contraindication for clopidogrel and patients infected with SARS-CoV-2 or other contagious respiratory diseases during patient recruitment. To facilitate study enrollment, pharmacists at each participating pharmacy identified patients eligible for PGx testing through their electronic pharmacy records. Patients with an active prescription for ticagrelor or prasugrel were contacted by phone and approached for study enrollment by their pharmacists. To encourage standardized study enrollment, all pharmacists were given an outline of talking points to utilize when patients were recruited. A pharmacist of every participating community pharmacy or pharmacy group and the treating cardiologist who prescribed the antiplatelet medication were invited to participate in a survey that elicited their perceptions on POC PGx testing in the community pharmacy and *CYP2C19*-guided de-escalation to clopidogrel.

### 2.5. Data Collection: POC CYP2C19 PGx Testing

CE-IVD-approved POC devices (Spartan) and ancillary test kits, intended for qualitative in vitro diagnostic test for the identification of a patient’s *CYP2C19* *2, *3, and *17 genotypes—determined from genomic DNA obtained from a buccal swab sample—were made available free of charge by the supplier (Angiocare, Amersfoort, the Netherlands). Genotype results were translated into *CYP2C19* metabolizer phenotypes according to the DPWG [28]. POC PGx testing in all locations was performed in the local pharmacy. All users received training in person by the supplier on using the POC device and buccal swabs. Since POC PGx testing took place during the COVID-19 pandemic (September 2021 and December 2021), patients were instructed to collect their own buccal swab. If self-collection was not possible, patients had the option of being swabbed by the provider in protective equipment. Upon written informed consent, patients underwent genotyping for *CYP2C19* *2,*3 and *17 according to the manufacturer’s guidelines. The turnaround time—defined as the time from pre-test counseling to the time the genotyping result was available—was logged by the community pharmacists. The local pharmacist had direct access to PGx screening results and was capable of recording them in the patient’s electronic patient record [29]. Pharmacists were educated by the investigators on *CYP2C19* genotype interpretation as well as corresponding phenotype designation as well as on the application of DPWG guidelines regarding the clopidogrel–*CYP2C19* drug–gene interaction. The prescribing cardiologist of patients eligible for clopidogrel de-escalation was approached by the pharmacists via phone or email to discuss the PGx test result and the implementation of the DPWG recommendations regarding this specific drug–gene interaction. Cardiologists were sent reminders until the remaining therapy duration undershot the minimum of three months, which was in accordance with the inclusion criteria. If the cardiologist agreed, patients without one or two *CYP2C19**2 or *3 alleles were de-escalated to clopidogrel according to the switching algorithm between oral P2Y_12_ inhibitors in the chronic setting of the current European Society of Cardiology (ESC) guidelines for dual antiplatelet therapy in coronary disease [30]. Because the *CYP2C19**17 allele only exhibits a modest increase in clopidogrel active metabolite formation, there are no significant implications for the *CYP2C19* phenotype and subsequent treatment recommendations for clopidogrel in comparison to the *CYP2C19**1 allele [17,28]. De-escalation to clopidogrel along with the *CYP2C19* test result was communicated with the patients by their pharmacist. Patients with a *CYP2C19* LOF-allele were maintained on ticagrelor or prasugrel therapy. All *CYP2C19* test results were communicated and explained to the patients in layman’s terms.

### 2.6. Data Collection: Patient, Pharmacist and Cardiologist Survey

Three surveys based on prior research on patient, pharmacist and physician experience with PGx were developed [31,32,33]. The patient questionnaire contained 16 close-ended and 4 open-ended questions on PGx testing in the community pharmacy, and answers were analyzed on a 5-point Likert scale ranging from 1 (strongly disagree) to 5 (strongly agree). The questionnaire was designed to cover the following themes: demographics, added value of PGx, attitude toward PGx testing in the community pharmacy, pharmacist–patient communication and sharing of PGx results (see Appendix A). The patient survey was conducted by telephone after the *CYP2C19* test results were communicated with the patients. As for the pharmacist survey, a telephone questionnaire was conducted to identify practical barriers and facilitators to implementation of POC PGx testing in the community pharmacy. The pharmacist survey gathered sociodemographic information and collected data on knowledge of PGx, attitude toward PGx testing in the community pharmacy, and pharmacist–patient and pharmacist–cardiologist communication (see Appendix A). Participating pharmacists were also asked to rank ten commonly perceived barriers to the implementation of PGx testing in community pharmacies based on an earlier exploratory study [32]. Lastly, the cardiologist survey covered questions on the provision of information about the *CYP2C19* test results in addition to the same themes found in the pharmacist survey (see Appendix A). Treating cardiologists had the opportunity to respond via e-mail or anonymously via Formdesk [34], which is an online survey tool.

A detailed patient and pharmacist journey and data collection is outlined in Figure 1.

### 2.7. Cost Analysis

We estimated average net costs per patient due to POC *CYP2C19* testing, which was presented in euros at price level 2023. Variables used to calculate cost savings per patient were de-escalation rate, therapy duration of P2Y_12_ inhibitor, percentage of patients on chronic P2Y_12_ inhibitor treatment, percentage of patients eligible for DAPT de-escalation, and ticagrelor-prasugrel ratio of study population. For costs of PGx testing, the supplier-provided list price of POC device and ancillary test kits was used. Drug costs per day were valued at €0.05 for clopidogrel (75 mg), €1.23 for prasugrel (5 mg), €2.38 for non-chronic ticagrelor (2 × 90 mg), and €2.48 for chronic ticagrelor (2 × 60 mg) [5]. Costs related to efficacy and safety events were excluded from the analysis due to the feasibility nature of our study. Costs per prevented major or minor bleeding according to the PLATO definition [35] were calculated based on the number of de-escalated patients needed to prevent one such bleeding. This number was set at 37, since the POPular Genetics Trial estimated that de-escalation reduces the PLATO major or minor bleeding risk from 12.5% to 9.8% [14]. The full spreadsheet model is presented in Appendix A Cost Analysis.

### 2.8. Data Analysis

All analyses were purely descriptive in nature, and no formal statistical calculations were performed due to the explorative nature of the study. Microsoft Excel was used to tabulate the results, and data were analyzed by using descriptive statistics. Explanatory comments of the survey were summarized.

## 3. Results

### 3.1. Participant Demographics

In total, 144 patients were enrolled in the study between September 2021 and December 2021. Patient characteristics are presented in Table 1. The average age of a patient in our study was 64 years (range: 34–87), 77% were male, and 97% self-reported Caucasian ethnicity. One hundred and twenty-seven patients (88%) received antiplatelet treatment with ticagrelor and seventeen patients (12%) received prasugrel. Median therapy duration at the time of enrolment was 5.9 months (range: 0.4–127). In five patients (3.5%), the remaining therapy duration was unclear, given their P2Y_12_ inhibitor duration at study inclusion (range: 9.5–11.2). One-fifth of the patients were on a chronic P2Y_12_ inhibitor regimen at the time of enrolment, which was defined as therapy duration >12 months.

A total of 27 community pharmacies comprised of both independent community pharmacies and community pharmacies that were part of a chain participated in the study. Fourteen pharmacists who are affiliated with the 27 participating community pharmacies participated in the pharmacist-specific survey. All pharmacy groups had at least one pharmacist represented in the survey. In addition, eight cardiologists of the patients eligible for therapy de-escalation to clopidogrel based on their *CYP2C19* genotype status participated in the survey tailored to cardiologists. Demographics of the cardiologists, pharmacists and their pharmacies are listed in Appendix A.

### 3.2. POC Genotyping Results

In total, 144 patients underwent *CYP2C19* POC genotyping in the community pharmacy. A pharmacy group of two community pharmacies (*n* = 37) and another group of five (*n* = 48) recruited the most patients and accounted for 59% of patient inclusion. Erroneous test results were observed in five patients (3.5%) at first attempt, two of whom were eventually excluded from the study: one patient withdrew from the study and another had two consecutive inconclusive results. Two patients had a successful test result on the second attempt, and after two inconclusive test results, the fifth patient underwent genotyping with the same POC device for the third time at the patient’s own request and eventually received a successful test result. Overall, 142 patients had a successful *CYP2C19* test result, resulting in a success rate of POC genotyping in the community pharmacy setting of 98.6%. Results of POC *CYP2C19* PGx testing and the average population statistics are presented in Table 2. The average turnaround time was approximately 75 min. This time included pre-test counseling, buccal swap sample collection, hands-on genotyping, and 1 h runtime of the *CYP2C19* POC device.

In total, out of the 142 patients successfully genotyped, 95 (67%) did not have a *CYP2C19* loss-of-function allele and were eligible to de-escalate to clopidogrel. Among these 95 patients without a *CYP2C19* LOF allele, 19 patients (20%) were de-escalated to clopidogrel after approval by the prescribing cardiologist. One patient was switched to a direct acting oral anticoagulant for unknown reasons. Patients de-escalated to clopidogrel had a median P2Y_12_ inhibitor therapy duration of 7.0 months (range: 1.5–89) at the time of enrollment. One (17%) of a total of six prasugrel patients eligible for therapy de-escalation switched to clopidogrel compared to eighteen (20%) of a total of eighty-nine ticagrelor patients eligible for de-escalation who switched to clopidogrel. The de-escalation rate varied widely among the participating pharmacy groups ranging from 60% (9/15) in one group of six community pharmacies to 0% (0/33) in another group of five pharmacies. Multiple pharmacies and pharmacy groups had a de-escalation rate of 0%. The de-escalation rate per pharmacy group is outlined in Appendix A. Of all 27 individual community pharmacies, the highest de-escalation rate seen in one community pharmacy was 75% (3/4).

The reasons indicated by cardiologists for not adopting the recommendation to switch to clopidogrel were documented by community pharmacists. The reason most patients were not de-escalated to clopidogrel was the perception of cardiologists being accountable when deviating from the local cardiology guidelines, which did not include PGx testing (*n* = 33, 35%). In 15 cases (16%), the treating cardiologist did not respond to the pharmacist’s recommendations. One cardiologist refrained from de-escalating to clopidogrel for the following reason: “These patients have almost all had (myocardial) infarctions, so that is a different indication (for therapy de-escalation)” (*n* = 10, 11%). Other reasons included unknown reason (*n* = 7, 7%), and therapy de-escalation not included in hospital guidelines (*n* = 5, 5%).

### 3.3. Patient Survey

The response rate to the patient survey was 84% (119/142). The response rate per item differed as not all respondents answered all questions.

#### 3.3.1. PGx in the Community Pharmacy

Almost all patients were positive about PGx testing in the community pharmacy setting when it comes to the convenience of PGx testing (*n* = 117, 100%) and the community pharmacy being a suitable location for PGx testing (*n* = 109, 92%). Of all respondents, 72 (61%) of the respondents stated that they would rather undergo PGx testing at the community pharmacy than at a laboratory or in a hospital. When asked to elaborate, the most common reported answers were the shorter distance to the pharmacy (*n* = 44, 61%) and shorter time to PGx testing (*n* = 13, 18%). Patient perceptions on PGx testing in a community pharmacy are outlined in Figure 2.

#### 3.3.2. Added Value of PGx

Among all responders to the survey, 103 (86%) indicated that they now understood how PGx testing is applied in healthcare, and 96 (81%) saw the benefits of PGx testing when using medication. In addition, more than half the patients (*n* = 63, 53%) reported to want pre-emptive PGx testing prior to initiation of drug therapy in the future. See Figure 3 for an overview of all patient survey answers on the added value of PGx. When patients were asked to comment on the positive aspects of the study, the most common reported answers were tailor-made pharmacotherapy (*n* = 29, 24%), better understanding of their medication (*n* = 24, 20%), and safety monitoring of drug therapy (*n* = 20, 17%). Regarding the negative aspects of the study, most patients (*n* = 99, 83%) reported there were none.

#### 3.3.3. Pharmacist-Patient Communication

The vast majority of respondents had confidence in the pharmacist when it comes to conducting and interpreting a PGx test (*n* = 107, 92%) and indicated that the information provided on potential changes to the antiplatelet drug was clear (*n* = 94, 85%). The PGx test was also explained in a way that was easy to understand, according to respondents (*n* = 103, 90%). All patient survey results on communication between the pharmacist and patient are presented in Appendix A.

#### 3.3.4. Sharing PGx Results

Most respondents were not concerned about who can see the results of the PGx test (*n* = 98, 82%). Almost all approved of the pharmacist having access to the PGx test result (*n* = 115, 97%) and would share test results with their doctors (*n* = 115, 97%). All patient survey results on sharing PGx results are shown in Appendix A.

### 3.4. Pharmacist Survey

A total of 14 pharmacists, representing all participating pharmacy groups, participated in the survey. The response rate was 100%. Interviewed pharmacists were on average 35 years old (range: 25–51), seven (50%) were managing pharmacists, and the average work experience was 9.4 years (range: 2–25). All pharmacists had prior experience with PGx test results and reported a mean PGx knowledge of 7 out of 10 (range: 5–8). The full demographics of participating pharmacists are presented in Appendix A. The most common reported barriers of POC PGx testing by the respondents were: poor accessibility and non-cooperation of treating cardiologists (*n* = 6, 43%), subpar knowledge and education of pharmacists (*n* = 6, 43%), and PGx testing being a difficult subject for patients (*n* = 5, 36%). Pharmacists reported that PGx in general being an unfamiliar field for cardiologists mainly contributed to their lack of cooperation. Other factors mentioned were PGx testing after the medication had already been started by the patient, the therapy duration of the P2Y_12_ inhibitor being twelve months in total instead of chronic, and no regular contact with the cardiologists from the hospital. As far as knowledge is concerned, not being educated enough about PGx and when it is of particular added value were reported as prominent factors affecting this barrier. Lastly, the unfamiliarity of patients with PGx made this subject difficult for them to fully understand. Pharmacists were also asked to rank ten well-known perceived barriers to the implementation of POC PGx testing in the community pharmacy. The ranking of these barriers is listed in Figure 4. A lack of reimbursement and costs associated with POC PGx testing were ranked highest, which was followed by limited time to commit to PGx testing and insufficient pharmacy staff to provide POC PGx testing. Respondents identified limited perceived added value to current practice and limited physical space in the pharmacy as the lowest ranked barriers, while lack of interest from patients and knowledge of PGx were also perceived as least significant barriers. Conversely, interviewed pharmacists also reported the most frequent facilitators: no resistance from patients (*n* = 7, 50%), PGx is the future (*n* = 5, 36%), and the added value of PGx (*n* = 4, 29%). Pharmacists stated that they felt no resistance from patients and that almost all who were eligible for PGx testing were enthusiastic and wanted to participate in the study. The main reason the community pharmacists participated in this feasibility study was because they believed PGx testing is the future. All participating pharmacists had a positive attitude toward POC PGx testing in the community pharmacy, and nearly all respondents thought POC PGx in the community pharmacy has added value compared to PGx in the hospital and laboratory (*n* = 12, 86%); two pharmacists (14%) were neutral.

### 3.5. Cardiologist Survey

Eight cardiologists who were contacted for de-escalation to clopidogrel completed the questionnaire about their view on *CYP2C19* PGx testing. The response rate was 62% (8/13). Mean age of the respondents was 44 years (*n* = 7, range: 30–49). Most (*n* = 7, 88%) were cardiologist and medical specialists, and the average work experience was 13 years (*n* = 7, range: 4–22). See Appendix A for full demographics of the cardiologists. Almost all cardiologists (*n* = 7, 88%) believed PGx has added value, but three (38%) indicated they needed more assistance to put this into practice. The open-ended comments of the survey revealed several barriers and enablers of *CYP2C19* testing. Reported barriers included: no clear added value of PGx, insufficient knowledge about PGx, unsure if PGx results will influence prescribing policy, increased administrative burden if results are not automatically shown in electronic patient records, and the time associated with applying PGx in practice. Cardiologist-perceived enablers, on the other hand, included personalized medicine being beneficial and the willingness to reduce healthcare costs due to the use of ticagrelor and prasugrel.

### 3.6. Cost Analysis

Costs of the POC *CYP2C19* testing were estimated at €140 for the test materials, plus €10 for pharmacist time. The average impact on remaining treatment duration was estimated at 7 months for standard 12-month users and 26 months for chronic users. In the base-case scenario, representing our study conditions, savings on medication resulted in total net costs of €54 per patient (Table 3). When taking into account the probability of successful testing, eligibility for therapy de-escalation, and actual de-escalation, about one in eight tested patients (*n* = 19, 13%) actually had his or her medication changed. Additionally, a PLATO major or minor bleeding is estimated to be prevented in about one in three hundred patients. Costs would therefore be €17,000 per prevented PLATO major or minor bleeding. We also assessed various alternative scenarios that could improve the net costs (Table 3). Doubling the de-escalation rate to 40% would increase savings on medication and make POC *CYP2C19* testing cost saving. A 100% de-escalation rate would even result in €332 net savings per patient.

Increasing the duration of the treatment impact would also reduce costs. Eighty percent of our patients were non-chronic users with standard 12-month treatment duration. When they were tested in our study, five months of treatment duration had already passed. Testing these patients at the onset of treatment would increase the average treatment impact from 10.8 to 14.7 months, reducing the average net costs to €19 per patient. Similarly, among the 20% chronic users, if the base-case 2 year treatment impact would increase to 5 years or more, then POC *CYP2C19* testing would become cost saving.

Finally, testing could be limited to particular subsets of patients (alternative scenarios 5–9). Limiting testing to only non-chronic users would reduce the impact on treatment duration and increase average costs to €86 (or €43 with early testing). Alternatively, testing limited to only chronic users would be cost saving. In addition, as treatment with prasugrel is less expensive than ticagrelor, the net costs of POC *CYP2C19* testing are higher for prasugrel users than for ticagrelor users (€99 versus €48).

## 4. Discussion

To our knowledge, this is the first feasibility study of POC *CYP2C19* genetic testing by community pharmacists. Multiple other studies have evaluated pharmacist-led *CYP2C19* testing in community pharmacies. However, in these cases, buccal swabs were sent to an external laboratory for PGx analysis [24,25,36,37,38]. In this study, we evaluated various outcomes to investigate the feasibility of the implementation of POC PGx testing in the community pharmacy. Our study’s key findings indicate that: (1) although only 20% of all eligible patients were successfully de-escalated to clopidogrel, (2) community pharmacists were proficient in PGx testing, using a POC *CYP2C19* device and (3) such testing did not appear to be time consuming. For testing to be cost saving, (4) the de-escalation rate and/or duration of treatment impact would have to increase. Lastly, both (5) pharmacists and (6) patients had a positive attitude toward POC PGx testing in the community pharmacy. Our data therefore indicate that the genotype-guided treatment of oral P2Y_12_ inhibitors by community pharmacists using a point-of -care *CYP2C19* device is feasible.

Of all the patients included in the study, 98.6% obtained a successful *CYP2C19* test result and 96.5% on the first attempt, in spite of the fact that the vast majority collected their own DNA sample following the instructions of the pharmacists. The success rate of 96.5% on the first attempt is higher than seen in most other studies where *CYP2C19* POC genotyping is performed by trained personnel [26,39,40,41], higher than one study where this was unknown [42], and even higher than laboratory-based genotyping in one study [15]. The high success rate of PGx testing contributes significantly to the feasibility of POC PGx testing by community pharmacists, as it can preclude the collection of additional samples and reduce tests costs.

The mean turnaround time for *CYP2C19* POC genotyping by community pharmacists was approximately 75 min and included pre-test counseling, buccal swap sample collection, hands-on genotyping, and about 60 min runtime of the *CYP2C19* POC device. Fifteen minutes of pharmacist’s time is well in line with a review of PGx implementation studies reporting time spent on single-gene tests by pharmacists [43]. Compared to central laboratory-based genotyping and in-hospital genotyping, which particularly is of importance for hospital outpatient community pharmacies, *CYP2C19* POC genetic testing in the community pharmacy can reduce the average turnaround time considerably. One feasibility study of *CYP2C19* genotyping in the Netherlands, using data from the Popular Genetics study, reported turnaround times of 2:16 h and 52:32 h for, respectively, in-hospital genotyping, and central laboratory testing. [26] The short turnaround time impacts the feasibility of POC PGx testing by community pharmacists given that it enables quicker clinical decision making and saves the pharmacists time associated with the workflow complexities involved with in-hospital and laboratory-based PGx testing. According to the literature, a fast turnaround time through POC genotyping is imperative for both reactive and pre-emptive models of PGx testing in pharmacy practice [43].

In our study, 87 patients (61%) were extensive metabolizers (*1/*1, 1/*17) and eight patients (6%) were ultrarapid metabolizers (*17/*17) of *CYP2C19*. The results of POC PGx testing are well in line with population data [28] and the aforementioned Dutch POPular genetics study [14]. Of these 95 patients eligible for de-escalation, only 19 patients (20%) were successfully de-escalated to clopidogrel. The de-escalation rate varied considerably between pharmacy groups. Although our feasibility study was not designed to explain these findings, there are several factors that may account for the low acceptance rates of pharmacist recommendations by cardiologists. First, in this study, patients had been genotyped reactively for *CYP2C19* instead of pre-emptively to guide antiplatelet therapy. This is endorsed by the pharmacist survey respondents, who reported that along with the usual limited therapy duration of 12 months, reactive genotyping was among the possible factors contributing to the lack of cooperation and the low acceptance rate by treating cardiologists. However, conceptually, it can actually be beneficial to de-escalate to clopidogrel at a later stage of antiplatelet treatment. Studies have shown that there is an anti-ischemic benefit of the potent P2Y_12_ inhibitors ticagrelor and prasugrel over clopidogrel early when the risk of ischemic complications is highest [44,45]. Hemorrhagic events in contrast occur mainly in the maintenance part of treatment [46,47]. These findings are supported by multiple large-scale studies of guided de-escalation, by means of platelet function testing or PGx testing, and unguided de-escalation, which is described as ‘plain’ de-escalating, from ticagrelor or prasugrel to clopidogrel to optimize DAPT in patients undergoing PCI [14,21,48]. Furthermore, one-fifth of the patients included in our study appeared to be on a chronic P2Y_12_ inhibitor regimen, which demonstrates that de-escalation to clopidogrel at a later stage of the one-year DAPT treatment is also still feasible. The observation of patients being on a prolonged DAPT regimen in our study is no anomaly; both the DAPT study [49] and the PEGASUS-TIMI trial [50] showed a significant reduction in cardiovascular ischemic events with DAPT beyond 1 year after index PCI, albeit paired with an increased risk of bleeding, as compared with aspirin monotherapy. Hence, in patients who have tolerated DAPT without a bleeding complication, the current ESC guidelines recommend considering a prolonged DAPT regimen when the thrombotic risk is moderate to high and in the absence of an increased risk for major or life-threatening bleeding [51]. Second, a modest de-escalation rate, although not as low as 20%, was to be expected, since existing cardiology guidelines still support the universal use of ticagrelor or prasugrel over a genotype-guided selection of P2Y_12_ inhibitor [51]. According to our findings, the perception of cardiologists being accountable when deviating from local guidelines—which did not include PGx testing—was also responsible for most eligible patients not de-escalating to clopidogrel. In addition, current clinical practice guidelines offer no clear recommendations for DAPT de-escalation and merely offer to consider de-escalation in selected patients with acute coronary syndrome who are deemed unsuitable for maintained potent platelet inhibition [51] or when bleeding risk outweighs thrombotic risks [52]. However, a recent-published meta-analysis on guided DAPT, by either genetic testing or platelet function testing, showed a significant reduction in bleeding without the increase in ischemic events [53]. In addition, DAPT de-escalation is already a common phenomenon in clinical practice for several reasons: side effects, such as dyspnea in the case of ticagrelor, a perceived high bleeding risk, and for economic purposes [54,55,56]. A *CYP2C19*-guided approach to switch between P2Y_12_ inhibitors could thereby provide the clinician an effective and safe tool to optimize DAPT in these selected cases. Third, affiliated prescribing cardiologists have not been informed in advance about this study and were not educated on *CYP2C19* PGx testing to guide DAPT, since the de-escalation rate to clopidogrel was one of our feasibility outcomes. Among the barriers reported by the cardiologists in our survey included no clear added value of PGx, insufficient knowledge about PGx, and unsure if PGx results will influence prescribing policy. According to the literature, education in pharmacogenetics is imperative for cardiologist adoption of PGx testing in clinical practice [31,57,58,59] and can also significantly influence cardiologists’ attitude toward PGx testing, especially in case of clopidogrel [60]. A recent scoping review based on 43 studies on pharmacy practices incorporating PGx testing revealed that the cardiologists’ acceptance rate of recommendations by pharmacists after PGx testing was 61% when pharmacist education or expertise was mentioned as being part of the implementation model compared to 33% in studies without described pharmacist education [43]. Education included residency programs, seminars with exams, board certification, and e-learning. Since cardiologists also acknowledged the benefits of PGx and their willingness to reduce healthcare costs assigned to the use of ticagrelor and prasugrel, we speculate that the lack of knowledge of genotype-guided antiplatelet therapy de-escalation, and thus the absence of a transmural working agreement between the community pharmacies and the cardiology departments of the local hospitals also attributed to the low de-escalation rate in our study. Due to the explorative nature of our feasibility study, further research is required to test this hypothesis. Lastly, in addition to PGx screening, there are other clinical and technical variables influencing clinicians’ decision making regarding antiplatelet therapy selection. Such variables may include patient’s bleeding or ischemic risk, shortened or extended DAPT duration, history of stent thrombosis on antiplatelet treatment, occurrence of adverse events, comorbidities, co-medications, contra-indications, and technical aspects of the stent implanted (material, size, design, number of stents) [51,61].

In this study, we also outlined the community pharmacists’ perceived barriers preventing and facilitators enabling POC PGx testing in the community pharmacy. The participating pharmacists in our study reported poor accessibility and non-cooperation of treating cardiologists as a prominent barrier inhibiting the implementation and identified the unfamiliarity of cardiologists with PGx as the root cause. This is in accordance with the data of a Dutch PGx implementation study showing that the lack of knowledge and awareness of PGx of healthcare providers are, according to pharmacists, challenging the implementation of PGx testing in primary care [62]. As mentioned above, further research is required to investigate to what extent the absence of knowledge of cardiologists regarding *CYP2C19* PGx testing to guide DAPT contributes to their lack of adoption of the recommendations provided by the pharmacists. Additionally, not being educated enough about PGx was perceived as another strong barrier. A perceived low knowledge level of pharmacists on PGx is a well-documented barrier to implementing PGx testing in primary care [31]. Therefore, PGx testing by community pharmacists will likely require additional training in PGx. Lastly, even though pharmacists reported PGx as a difficult topic for patients, participating pharmacists also stated that almost all patients were enthusiastic about participating in the study. This was also reflected in our patient survey results. When pharmacists were asked to rank 10 well-known barriers to the implementation of PGx testing in the community pharmacy setting, the top responses provided were costs of PGx testing and lack of reimbursement. In the Netherlands, there is no reimbursement for pre-emptive PGx testing or PGx testing for guided de-escalation of antiplatelet therapy. In addition, the time required for a pharmacist to counsel the patient and to consult with the treating cardiologist is not reimbursed. However, in our study, all participating pharmacists had a positive attitude toward POC PGx testing in the community pharmacy, and nearly all (86%) believed in the added value of PGx testing in the community pharmacy compared to in-hospital and central laboratory-based genotyping.

With regard to the patient survey, more than 80% of patients consented, which is remarkable since only 20% of all study patients were successfully de-escalated to clopidogrel based on their *CYP2C19* test results. While the de-escalation rate was low, patients were positive about PGx testing when using medication. This is as anticipated, given the fact that the patient’s strong interest in PGx has been well documented [31,63,64]. The absence of expenditures associated with PGx testing in our study, and the fact that it was non-invasive, may also have contributed to the patients’ attitude toward POC PGx testing. Nearly all patients found the community pharmacy a suitable location to undergo PGx testing, and the majority preferred POC genotyping in the community pharmacy over in-hospital and central laboratory-based PGx testing. In this regard, patient perceptions are in line with similar screening services offered by community pharmacists [65,66].

Our cost analysis indicated that the costs per prevented PLATO major or minor bleeding in the base-case scenario was estimated at €17,000, which may be considered high since the primary benefit of genotype-guided de-escalation to clopidogrel is a reduced incidence of minor bleeding [14]. However, it is essential to emphasize that even PLATO minor bleeding entails medical intervention and can impact healthcare costs [35]. For testing to be cost saving, the de-escalation rate needs to improve (from 20% to at least 40%), and/or the duration of treatment impact would have to increase (for example by testing earlier or testing only chronic users). Earlier studies on genotype-guided P2Y_12_ inhibitor therapy selection showed cost-effectiveness based on the premise of a 100% de-escalation rate [67,68,69,70]. This assumption has been deemed unrealistic, but our study shows that less than 100% de-escalation can still suffice to obtain cost savings.

Whilst recognizing the feasibility nature of this study, some limitations should be considered. First, the sample size was modest, and the results may not be generalizable to other community pharmacies in the Netherlands, as external validity may be low. Second, genetic tests are classified as high complexity tests by the FDA and therefore are required to be performed by a Clinical Laboratory Improvement Amendments (CLIA) laboratory, prohibiting their use as POC tests in the United States [71]. Third, the study patients were drawn from areas with limited geographic diversity, and almost all patients were of self-identified Caucasian descent. The prevalence of *CYP2C19* LOF alleles in this population is much lower than in Asian populations [72]. As a consequence, the ratio of ticagrelor or prasugrel patients eligible for de-escalation to clopidogrel in this demographic group will be lower, which in turn may affect feasibility outcomes. Fourth, given the feasibility nature of this study, no comparative analyses were possible to discriminate between the reasons affecting the low de-escalation rate. Fifth, the patient survey results may be subject to participation bias, as the views of patients who declined to participate in the study were not captured, potentially affecting the generalizability of these findings. Sixth, for the cost analysis modeling, assumptions needed to be made, for example that savings on PLATO major or minor bleeding costs could be ignored. Lastly, as with many other *CYP2C19* POC devices, our POC device only identifies best characterized variants such as *CYP2C19* *2, *3, and *17 genotype, and it may as a consequence label other rare *CYP2C19* LOF-alleles (*CYP2C19**4, *5, *6, *7, *8) as wild type-like. However, the prevalence of these genotype variants is low [72] and would therefore not impact our feasibility outcomes.

Despite these limitations, our study also has several strengths. It is the first feasibility study of PGx sampling and analysis in a community pharmacy setting. The ability to document PGx test results in the electronic patient records of community pharmacies in the Netherlands and to share them with other community and outpatient pharmacies, general practitioner practices, and hospitals enables healthcare providers to recall results when future drug–gene interactions are encountered [27,73,74]. Furthermore, the feasibility of POC PGx testing in the outpatient setting was investigated by genotyping for *CYP2C19*, which is a highly actionable PGx test: two-thirds of the population on prasugrel or ticagrelor therapy are eligible for DAPT de-escalation, and identifying these patients can reduce both bleeding events and drug costs. Moreover, the perspectives of all relevant stakeholders in the implementation of POC PGx testing were included by gathering survey data from patients, pharmacists and cardiologists. Ultimately, this study serves as a pilot endeavor toward community pharmacists to optimize antiplatelet therapy in patients on a P2Y_12_ inhibitor regime with either ticagrelor or prasugrel based on their *CYP2C19*-genotype status. Going forward, studies could apply this proof-of-concept model to other settings to facilitate the widespread implementation of personalized medicine, by means of PGx testing, in primary care.

## 5. Conclusions

Our findings demonstrate the feasibility of using POC PGx testing by community pharmacists for *CYP2C19*-guided de-escalation to clopidogrel. This approach has the potential to be cost saving and can be further amplified through the education of healthcare providers on genotype-guided DAPT and the wider adoption of *CYP2C19* PGX testing in cardiology guidelines.

## Figures and Tables

**Figure 1 genes-14-00578-f001:**
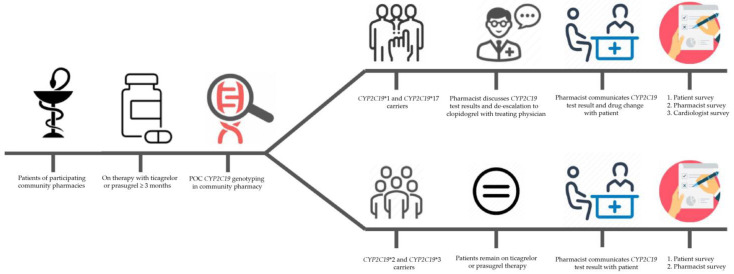
Patient and pharmacist journey per study patient undergoing POC (POC) *CYP2C19* testing in the community pharmacy. Patients of participating community pharmacies with an indication for ticagrelor or prasugrel for at least three months are recruited by their community pharmacist. After informed consent, patients are genotyped for *CYP2C19* *2, *3, and *17 in the community pharmacy. Patients with only *CYP2C19**1 and *17 allele are eligible for antiplatelet therapy de-escalation and switched to clopidogrel in consultation with treating cardiologist. Upon cardiologist’s approval, pharmacists communicate therapy de-escalation to clopidogrel with patients. Patients with a loss-of-function (LOF) *CYP2C19* *2 or *3 polymorphism remain on ticagrelor or prasugrel antiplatelet therapy. All *CYP2C19* test results are communicated with the patients by the pharmacist. All (1) patients with a *CYP2C19* test result, (2) participating pharmacists, and (3) treating cardiologists- who have been approached by pharmacists for de-escalation to clopidogrel- are invited to participate in a general survey on POC *CYP2C19* testing in the community pharmacy.

**Figure 2 genes-14-00578-f002:**
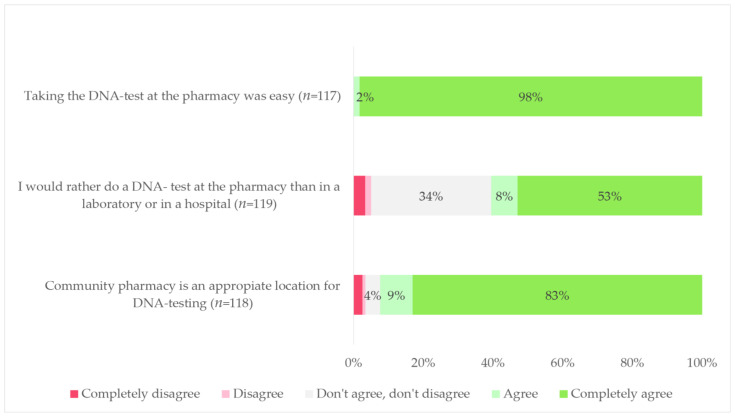
Patient survey results on pharmacogenetic testing in community pharmacy. DNA test: pharmacogenetic test. Percentages may not add to 100% because of rounding.

**Figure 3 genes-14-00578-f003:**
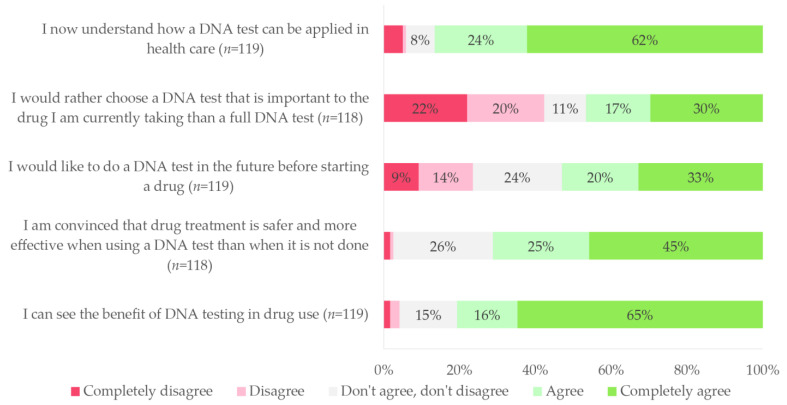
Patient survey results on the added value of pharmacogenetic testing. DNA test: pharmacogenetic test. Percentages may not add to 100% because of rounding.

**Figure 4 genes-14-00578-f004:**
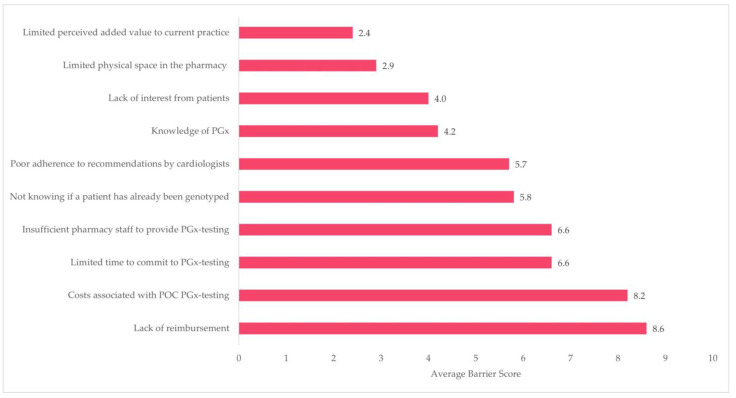
Community pharmacists’ ranking of ten commonly perceived barriers [32] to the implementation of PGx testing in community pharmacies. The average barrier score is the average score from 0 to 10 given to each barrier by each community pharmacist (*n* = 14). The barrier with the highest score is ranked first, while the barrier with the lowest score value is ranked least. Matrix ranking and barrier scores given by each community pharmacists are presented in Appendix A. PGx, pharmacogenetics; POC, point-of-care.

**Table 1 genes-14-00578-t001:** Characteristics of patients enrolled in the study.

Category	Value
Number of patients	144
Age (mean, range)	64, 34–87 years
Gender	
Male	110 (76%)
Female	34 (24%)
P2Y_12_ inhibitor regimen	
Ticagrelor	127 (88%)
Prasugrel	17 (12%)
P2Y_12_ therapy duration at study inclusion	
All users (median, range)	5.9 (0.4–127) months
12-month users (mean, range)	4.8 (0.4–11.2) months
Chronic users (mean, range)	38 (13–127) months
Patients on chronic treatment (>12 months)	29 (20%)

**Table 2 genes-14-00578-t002:** Results of *CYP2C19* POC testing.

*CYP2C19* Genotype	*CYP2C19* Phenotype	Study Sample (%)	Expected Percentages from Population Demographics (%)
*1/*1, 1/*17	Extensive metabolizer	87/142 (61%)	61–71
*1/*2, *1/*3, *2/*17, *3/*17	Intermediate metabolizer	41/142 (29%)	23–30
*17/*17	Ultrarapid metabolizer	8/142 (5.6%)	4.8–5.8
*2/*2, *2/*3, *3/*3	Poor metabolizer	6/142 (4.2%)	1.7–3.3

Percentages may not add to 100% because of rounding. Expected percentages from population demographics taken from [14,28]. *CYP2C19* genotypes were translated into phenotypes according to [28].

**Table 3 genes-14-00578-t003:** Average costs due to POC PGx testing, depending on key cost drivers.

	Key Cost Drivers			Outcomes	
	De-Escalation Rate	Average Duration of Treatment Impact (in Months)	Use of Ticagrelor versus Prasugrel	Costs per Patient (in €)	Costs per Prevented PLATO Major or Minor Bleeding
**Base-case scenario**	20%	10.8	88%	54	€17,000
**Alternative scenarios**					
1. Improved de-escalation	40%	*	*	−43	Cost saving
2. Full de-escalation	100%	*	*	−332	Cost saving
3. Early testing in standard 12-month users	*	14.7	*	19	€4000
4. Five-year impact in chronic users	*	17.7	*	−7	Cost saving
5. Testing only in standard 12-month users	*	7.2	*	86	€40,000
6. Testing only and early in standard 12-month users	*	12.0	*	43	€12,000
7. Testing only in chronic users	*	25.6	*	−77	Cost saving
8. Testing only in prasugrel users	*	*	0%	99	€31,000
9. Testing only in ticagrelor users	*	*	100%	48	€15,000

* For the alternative scenarios, assumptions are unchanged unless indicated otherwise.

## Data Availability

The data presented in this study are available in article and Appendix A.

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
