# Peer review of "Feasibility of Community Pharmacist-Initiated and Point-of-Care CYP2C19 Genotype-Guided De-Escalation of Oral P2Y12 Inhibitors"

_genes, 2023, doi:10.3390/genes14030578_

Round 1

Reviewer 1 Report

Overall this is a well-written manuscript about the exploratory use of clinical testing in the pharmacy for CYP2C19 testing.  

My major concern is the use of the term point of care testing.  While this study was performed in the EU (Netherlands) where the point of care device is define as any analytical test performed for a patient by a healthcare professional outside the conventional laboratory setting.  In the US, although the FDA granted the Spartan Rx CYP2C19 System 510 (k) clearance, Spartan did not satisfy the agency's requirements to garner clearance for the test with a point-of-care indication.  I think this needs to be clear in the manuscript as there is significant confusion related to the Spartan test system.  

The other concern that the authors address relatively well is that pharmacists do not have the ability to prescribe meds.  So while testing was performed in the pharmacy, if the physician doesn't change the meds then testing in the pharmacy doesn't make sense.  

I also would like to see described a little better how the pharmacy results were incorporated into the medical record, if there were at all.  

Most of the people that participated took part of the surveys.  Did any of the people who did not participate take the surveys on patient satisfaction?  I suspect that there could be a bias.  It might be good to mention that possibility.  

Minor comments:

I tend not to like the term carrier/carrying in pharmacogenetics.  Carrier/Carrying comes from autosomal recessive disorders in genetics where having 1 copy of a pathogenic variant does not confer disease.  In PGx, one loss of function variant can impact dosing dependent on the drug-gene pair, especially for clopidogrel and CYP2C19.  I recommend using the term have/having or something similar.  

page 3, line 139 - typo, please change manfacture's to manufacturer's

Table 2 - according to CPIC *1/*17 is a rapid metabolizer.  Though for clopidogrel rapid metabolizers are dosed the same as normal metabolizers.  Please clarify.  

Author Response

Dear editor,

Thank you for providing us the opportunity to revise our manuscript entitled ‘Feasibility of community pharmacist-initiated and point-of-care CYP2C19-guided de-escalation of oral P2Y12 Inhibitors’ which has been assigned the reference number 2235591. We appreciate the comments and feedback provided by the reviewers, and have made significant changes to improve the quality and clarity of the manuscript.

The following is a point-by-point response to referee 1:

  1. Overall this is a well-written manuscript about the exploratory use of clinical testing in the pharmacy for CYP2C19 testing.
    1. Thank you very much.
  2. My major concern is the use of the term point of care testing. While this study was performed in the EU (Netherlands) where the point of care device is define as any analytical test performed for a patient by a healthcare professional outside the conventional laboratory setting.  In the US, although the FDA granted the Spartan Rx CYP2C19 System 510 (k) clearance, Spartan did not satisfy the agency's requirements to garner clearance for the test with a point-of-care indication.  I think this needs to be clear in the manuscript as there is significant confusion related to the Spartan test system. 
    1. To address this comment we have added that genetic tests can only be performed by a CLIA laboratory in the US in the limitations part of the discussion.
  3. The other concern that the authors address relatively well is that pharmacists do not have the ability to prescribe meds. So while testing was performed in the pharmacy, if the physician doesn't change the meds then testing in the pharmacy doesn't make sense.
    1. Thank you for your kind consideration. We have explained this in the paper and agree that it is the physician that ultimately decides to change the prescription as he is the only one who oversees all factors that are important for selection of the right treatment. We disagree that given this fact it does not make sense to perform the test in the pharmacy as pharmacist are widely recognized as PGx experts and being able to execute the test in the pharmacy is complimentary to their knowledge.
  4. I also would like to see described a little better how the pharmacy results were incorporated into the medical record, if there were at all.
    1. This is now described in 5. data collection: POC CYP2C19 PGx testing. All pharmacists had the ability to record the PGx results in the electronic medical record. A (Dutch) reference is also added to support this statement
  5. Most of the people that participated took part of the surveys. Did any of the people who did not participate take the surveys on patient satisfaction?  I suspect that there could be a bias.  It might be good to mention that possibility. 
    1. Unfortunately no data have been collected from patients who refused to participate in the study. To address the potential bias resulting from this we have mentioned this in the limitations part of the discussion.
  6. I tend not to like the term carrier/carrying in pharmacogenetics. Carrier/Carrying comes from autosomal recessive disorders in genetics where having 1 copy of a pathogenic variant does not confer disease.  In PGx, one loss of function variant can impact dosing dependent on the drug-gene pair, especially for clopidogrel and CYP2C19.  I recommend using the term have/having or something similar.
    1. In line with the reviewers comments we do not use the term ‘carry’ (e.g. LOF alleles) anymore.
  7. page 3, line 139 - typo, please change manfacture's to manufacturer's
    1. We have changed this, thank you.
  8. Table 2 - according to CPIC *1/*17 is a rapid metabolizer. Though for clopidogrel rapid metabolizers are dosed the same as normal metabolizers.  Please clarify. 
    1. The impact of CYP2C19*17 allele in relation to CYP2C19*1 allele is now mentioned in ‘2.5 data collection: POC CYP2C19 PGx testing’.

Furthermore, we removed the low de-escalation rate as a limitation of the study, as this is a result of the study, and is already mentioned in the discussion and conclusion.

All changes have been marked with track changes in Word, making it easy for you and the reviewers to see the modifications that were made. Overall, we strongly believe that these changes have significantly strengthened the manuscript and addressed the concerns raised by the reviewers. Thank you again for your guidance and support throughout this process.

Sincerely,

Amar Levens

Reviewer 2 Report

This is an important study by Levens and colleagues on the potential for PGx testing in the pharmacy setting.  The study is well described and carefully carried out.  The results are very interesting and will be well received by the PGx international community.  Limitations have been acknowledged but won't detract from the interpretation. The only suggestion for improvement would be to improve the readability of the text in all of the figures.

Author Response

Dear editor,

Thank you for providing us the opportunity to revise our manuscript entitled ‘Feasibility of community pharmacist-initiated and point-of-care CYP2C19-guided de-escalation of oral P2Y12 Inhibitors’ which has been assigned the reference number 2235591. We appreciate the comments and feedback provided by the reviewers, and have made significant changes to improve the quality and clarity of the manuscript.

The following is a point-by-point response to referee 2:

  1. This is an important study by Levens and colleagues on the potential for PGx testing in the pharmacy setting. The study is well described and carefully carried out.  The results are very interesting and will be well received by the PGx international community.  Limitations have been acknowledged but won't detract from the interpretation. The only suggestion for improvement would be to improve the readability of the text in all of the figures.
    1. Thank you very much for your kind words and for your careful review of our manuscript. We are glad that you found our study to be important and well-described, and we appreciate your feedback about improving the readability of the figures. We have taken your suggestion to heart and have done our best to improve the text in all of the figures to make them as clear and accessible as possible. Thank you again for your valuable feedback and for your support of our research.

Sincerely,

Amar Levens